# ADAPTIVE POLICY BACKBONE VIA SHARED NETWORK

## ABSTRACT

Reinforcement learning (RL) has achieved impressive results across domains, yet learning an optimal policy typically requires extensive interaction data, limiting practical deployment. A common remedy is to leverage priors—such as pre-collected datasets or reference policies—but their utility degrades under task mismatch between training and deployment. While prior work has sought to address this mismatch, it has largely been restricted to in-distribution settings. To address this challenge, we propose **A**daptive **P**olicy **B**ackbone (APB), a meta-transfer RL method that inserts lightweight linear layers before and after a shared backbone, thereby enabling parameter-efficient fine-tuning (PEFT) while preserving prior knowledge during adaptation. Our results show that APB improves sample efficiency over standard RL and adapts to out-of-distribution (OOD) tasks where existing meta-RL baselines typically fail.

## 1 INTRODUCTION

Reinforcement learning (RL) has demonstrated impressive ability to learn high-performing policies across diverse domains, including gaming (Mnih et al., 2015; Vinyals et al., 2019; Berner et al., 2019), robotics (Levine et al., 2016; Akkaya et al., 2019), traffic control (El-Tantawy et al., 2013), and aligning large language models with human preferences (Stiennon et al., 2020). However, learning a high-performing policy in RL typically requires extensive data collection, thereby discouraging practical deployment.

To alleviate the burden of extensive data collection, recent work has explored leveraging *priors*, either via a pre-collected dataset (Levine et al., 2020) or a reference policy (Xie et al., 2021; Kalashnikov et al., 2018). In practice, however, the deployment task often differs from that represented by the dataset or reference policy; such *task mismatch* can substantially diminish the utility of these priors. To leverage priors despite this mismatch, several approaches have been proposed in the context of meta-RL (Wang et al., 2016; Duan et al., 2016; Finn et al., 2017; Rakelly et al., 2019), which aim to leverage prior knowledge for efficient adaptation, either by (i) improving the sample efficiency of standard RL or by (ii) enabling rapid adaptation to new tasks.

Despite progress on handling task mismatch, many methods still struggle with adapting to out-of-distribution (OOD) tasks and thus fail to leverage prior knowledge, because they implicitly assume that training and deployment tasks are drawn from the same distribution, which is rarely realistic. For example, Finn et al. (2017) evaluates policy adaptation on HalfCheetah-vel, where training tasks are reaching target velocities uniformly sampled from $[0, 3]$ and, after training, the policy is adapted to new targets drawn from the same range. However, one typically desires methods that adapt effectively even when tasked with moving backward (negative target velocity), which is fundamentally OOD. This gap highlights the need for methods that reliably adapt to tasks beyond those encountered during training.

In this paper, we propose the **A**daptive **P**olicy **B**ackbone (APB), a meta-transfer RL method for sample-efficient adaptation on OOD tasks. To address the challenges posed by OOD tasks, we adopt an initialization-based meta-RL paradigm that learns a meta-initialization and adapts via fine-tuning at test time (Beck et al., 2023). As illustrated in Figure 1, APB consists of a backbone shared across meta-training tasks, together with task-specific linear layers placed before and after it. After meta-training the backbone, APB employs parameter-efficient fine-tuning (PEFT), enabling adaptation by updating only the task-specific linear layers while preserving previously acquired knowledge in

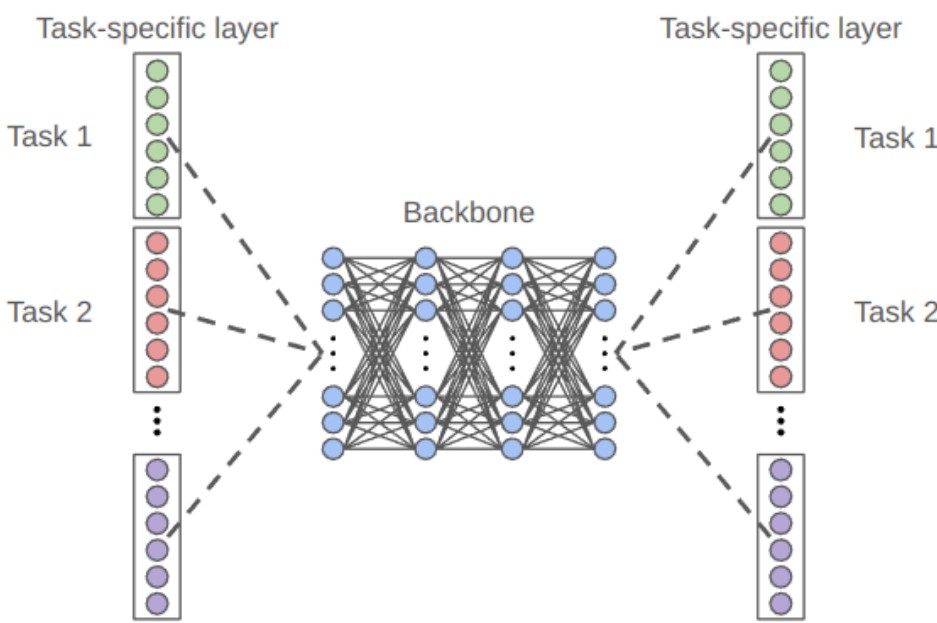

Figure 1: APB architecture: a shared backbone with task-specific linear layers.

the backbone. Prior work has investigated PEFT for meta-RL and has reported advantages over full fine-tuning, but primarily within in-distribution settings (Zintgraf et al., 2019a; Raghu et al., 2019). In contrast, we focus on OOD task adaptation and provide a theoretical analysis.

Our main claims are as follows. First, training only the pre- and post-backbone linear layers while maintaining the backbone parameters is sufficient to adapt to the new task. To support our claim, we present a theoretical analysis under a simple setting and provide empirical results on more complex environments. The results show that APB can improve the sample efficiency over the standard RL algorithms and adapts to OOD tasks where existing meta-RL methods typically fail. Second, APB is capable of generalizing behavior on OOD tasks. To validate the generalization capability of APB, we conduct a behavior cloning (BC) evaluation with OOD expert demonstrations whose trajectory distribution is narrowly concentrated, a regime in which vanilla BC typically fails to generalize behavior. From the perspective of leveraging backbone parameters trained on meta-training tasks for enhanced sample efficiency, APB can be interpreted as transfer learning, while it may also be regarded as meta-learning in terms of improving generalization to OOD tasks.

## 2 RELATED WORKS

**Parameter-Efficient Fine-Tuning.** Fine-tuning is a widely used approach to adapt pre-trained models to downstream tasks by adjusting model parameters. However, full fine-tuning can be inefficient and often leads to catastrophic forgetting, where previously learned knowledge is overwritten and performance deteriorates. Recently, parameter-efficient fine-tuning (PEFT) has emerged as a promising alternative. PEFT enables adaptation to downstream tasks while mitigating catastrophic forgetting, updating only a small subset of parameters and freezing the rest. Previously, PEFT has been implemented by making only adapter parameters learnable (Zhang et al., 2023; Hu et al., 2022; Sung et al., 2022; Houlsby et al., 2019), or by updating only the bias terms (Zaken et al., 2021), a single layer (Lee et al., 2022; Zhu et al., 2023; Kumar et al., 2022), or a layer block (Peng et al., 2024). Due to its effectiveness on downstream tasks, PEFT is utilized in various areas such as LLM (Fu et al., 2023; Zhang et al., 2023; Hu et al., 2022; Houlsby et al., 2019; Zaken et al., 2021; Zhu et al., 2023), vision (Lee et al., 2022; Sung et al., 2022; Kumar et al., 2022; Peng et al., 2024; Jia et al., 2022; Jie & Deng, 2023; Wang et al., 2023), robot learning for sim2real (Truong et al., 2021; Sharma et al., 2023), and meta-RL (Zintgraf et al., 2019a; Raghu et al., 2019).

**Transfer Learning.** Transfer learning (TL) is an approach for transferring knowledge learned in a source domain to accelerate learning in a target domain. Because TL aims to transfer informative representations that improve generalization, these ideas have been applied to meta-RL (Devin et al., 2017; Borsa et al., 2018; Zintgraf et al., 2019a; Raghu et al., 2019). Devin et al. (2017) pre-train modular policy components across tasks and dynamically reassemble them for new tasks. Zintgraf et al. (2019a); Raghu et al. (2019) pre-train policies and adapt by fine-tuning only a subset of parameters to enable fast adaptation. Borsa et al. (2018) leverage successor features to promote generalization across tasks. However, successor-feature methods can degrade under drastic changes in the reward function across domains, and they typically assume shared transition dynamics (Lehnert et al., 2017; Borsa et al., 2018).

**Meta Learning.** Meta-learning is a subfield of machine learning that aims to train models across a variety of tasks such that they can rapidly solve new, unseen tasks. The objectives of meta-learning include learning effective parameter initializations (Finn et al., 2017; Ravi & Larochelle, 2017; Nichol et al., 2018), discovering task-specific update rules or learning algorithms (Santoro et al., 2016; Munkhdalai & Yu, 2017; Mishra et al., 2017), and learning task similarity metrics (Koch et al., 2015; Vinyals et al., 2016; Snell et al., 2017; Sung et al., 2018).

Meta-learning has recently been actively investigated in the context of RL, known as meta-RL. Wang et al. (2016); Duan et al. (2016) propose recurrent approaches that encode experience to infer task-specific dynamics. Finn et al. (2017) introduces Model-Agnostic Meta-Learning (MAML), which aims to learn an initial set of parameters that can be quickly adapted to other tasks with a few gradient steps. Several variants of MAML have subsequently been proposed (Nichol et al., 2018; Stadie et al., 2018; Zintgraf et al., 2019a; Raghu et al., 2019; Song et al., 2019). Meanwhile, Rakelly et al. (2019); Zintgraf et al. (2019b); Beukman et al. (2023); Liang et al. (2024) employ a context encoder to infer task identity from transition history, enabling efficient posterior inference for adaptation. Recently, to improve generalization over task distributions, Xu et al. (2022); Schmied et al. (2023); Wang et al. (2024) utilize the transformer-based architecture (Vaswani et al., 2017).

## 3 PRELIMINARIES

### 3.1 PROBLEM FORMULATION

We consider a Markov decision process (MDP) defined by the tuple $(\mathcal{S}, \mathcal{A}, P, r, \gamma)$. $\mathcal{S}$ is the state space, $\mathcal{A}$ is the action space, $P(\cdot|s,a)$ is the transition kernel over $\mathcal{S}$, $r : \mathcal{S} \times \mathcal{A} \to [r_{\min}, r_{\max}]$ is the reward function, and $\gamma \in (0,1)$ is a discount factor that controls the weighting of future rewards. At each time step $t$, given state $s_t \in \mathcal{S}$, the agent chooses an action $a_t \in \mathcal{A}$ according to a (possibly stochastic) policy $\pi(a|s)$. The state then transitions from $s_t$ to $s_{t+1}$ according to $P(s_{t+1}|s_t, a_t)$, and the agent receives a reward $r(s_t, a_t)$. Moreover, the state-value function under policy $\pi$ is defined as

$$v^\pi(s_t) = \mathbb{E}_{\substack{a_k \sim \pi(\cdot|s_k) \\ s_{k+1} \sim P(\cdot|s_k, a_k)}} \left[ \sum_{k=t}^{\infty} \gamma^{k-t} r(s_k, a_k) \;\middle|\; s_t \right]$$

The objective of RL is to learn a policy $\pi$ that maximizes the expected return $\mathbb{E}_{s_0 \sim \rho_0}[v^\pi(s_0)]$, where $\rho_0$ denotes the initial-state distribution.

Here, we formalize the meta-RL setting as follows. Let $p(\mathcal{T})$ denote a distribution over tasks, where each task $\mathcal{T}_i \sim p(\mathcal{T})$ corresponds to an MDP $(\mathcal{S}, \mathcal{A}, P, r_\mathcal{T}, \gamma)$. In this work, task variation arises from differences in reward functions. During meta-training, we sample $N$ number of tasks $\{\mathcal{T}_i\}_{i=1}^N \sim p(\mathcal{T})$, and interact with these tasks to learn shared parameters. At evaluation, a novel task $\mathcal{T}_o \sim q(\mathcal{T})$ is drawn from a distribution $q(\mathcal{T})$ that differs from $p(\mathcal{T})$ (i.e., OOD with respect to $p(\mathcal{T})$; e.g., $\mathrm{supp}(q) \nsubseteq \mathrm{supp}(p)$).

### 3.2 MATRIX EXPRESSION

Although the MDP considered in this paper has continuous spaces, we adopt a discrete setting for matrix-based analysis and slightly overload notation to denote matrix representation. Under policy $\pi$, define $V^\pi$, $R^\pi$ and $P^\pi$ as the state-value vector, the expected reward vector, and the state

transition matrix, respectively, presented as follows:

$$V^\pi = \begin{bmatrix} v^\pi(s_1) \\ \vdots \\ v^\pi(s_{|\mathcal{S}|}) \end{bmatrix} \in \mathbb{R}^{|\mathcal{S}|\times 1}, R^\pi = \begin{bmatrix} \mathbb{E}_{a\sim\pi(\cdot|s_1)}\left[r(s_1,a)\right] \\ \vdots \\ \mathbb{E}_{a\sim\pi(\cdot|s_{|\mathcal{S}|})}\left[r(s_{|\mathcal{S}|},a)\right] \end{bmatrix} \in \mathbb{R}^{|\mathcal{S}|\times 1}$$

$$P^\pi = \begin{bmatrix} \mathbb{E}_{a\sim\pi(\cdot|s_1)}\left[P(s_1|s_1,a)\right] & \cdots & \mathbb{E}_{a\sim\pi(\cdot|s_1)}\left[P(s_{|\mathcal{S}|}|s_1,a)\right] \\ \vdots & \ddots & \vdots \\ \mathbb{E}_{a\sim\pi(\cdot|s_{|\mathcal{S}|})}\left[P(s_1|s_{|\mathcal{S}|},a)\right] & \cdots & \mathbb{E}_{a\sim\pi(\cdot|s_{|\mathcal{S}|})}\left[P(s_{|\mathcal{S}|}|s_{|\mathcal{S}|},a)\right] \end{bmatrix} \in \mathbb{R}^{|\mathcal{S}|\times|\mathcal{S}|}$$

Assuming sufficiently long or infinite-horizon episodes, $V^\pi$ can be expressed as:

$$V^\pi = (I_{|\mathcal{S}|} - \gamma P^\pi)^{-1} R^\pi$$

where $I_{|\mathcal{S}|} \in \mathbb{R}^{|\mathcal{S}|\times|\mathcal{S}|}$ is the identity matrix. Following Wang et al. (2007); Luan et al. (2019); Lim & Lee (2024), let $\Pi \in \mathbb{R}^{|\mathcal{S}|\times|\mathcal{S}||\mathcal{A}|}$ be the policy matrix with $(\Pi)_{s,(s',a)} = \pi(a \mid s)\mathbf{1}\{s' = s\}$ and the $\boldsymbol{r} \in \mathbb{R}^{|\mathcal{S}||\mathcal{A}|\times 1}$ the state-action reward vector with $\boldsymbol{r}_{(s,a)} = r(s,a)$. Then

$$P^\pi = \Pi P, \quad R^\pi = \Pi\boldsymbol{r}$$

## 4 ADAPTIVE POLICY BACKBONE

We propose Adaptive Policy Backbone (APB), a simple yet effective meta-transfer RL method designed to improve sample efficiency over standard RL and to enable adaptation to OOD tasks where existing meta-RL methods often fail. We argue that updating only the first and last layers of the policy network—while freezing the backbone to preserve prior knowledge—suffices for effective adaptation to new tasks. The key intuition is that although each optimal policy is task-specific, policies share a common component induced by structural similarities in the underlying MDPs; this shared structure reduces the need for extensive retraining on new tasks. Furthermore, it is well-established that fine-tuning only a subset of parameters can significantly improve sample efficiency and reduce training costs. Structurally, APB is analogous to D'Eramo et al. (2024), which uses a backbone shared across training tasks together with task-specific nonlinear modules placed before and after the backbone. While that line of work emphasizes how shared knowledge aids per-task training, APB replaces the non-linear modules with linear layers for greater parameter efficiency and targets test-time adaptation to OOD tasks in a meta-learning setting. To support our main claim, we provide a theoretical analysis of the policy structure in a simple environment, and extend our study to more complex environments through empirical experiments.

### 4.1 THEORETICAL STUDY OF THE POLICY STRUCTURE

We begin our analysis with a simple environment with finite state and action spaces.

**Lemma 1.** *If $V^\pi = (I_{|\mathcal{S}|} - \gamma P^\pi)^{-1} R^\pi$ holds, then using $P^\pi = \Pi P$ and $R^\pi = \Pi\mathbf{r}$ we have $V^\pi = \Pi(\gamma PV^\pi + \mathbf{r})$. Consequently, the policy matrix $\Pi$ can be expressed as*

$$\Pi = \frac{V^\pi\left(\gamma PV^\pi + \mathbf{r}\right)^\top}{\left\|\gamma PV^\pi + \mathbf{r}\right\|^2} + N \quad \text{with} \quad N(\gamma PV^\pi + \mathbf{r}) = \mathbf{0}_{|\mathcal{S}|} \tag{1}$$

*where $N \in \mathbb{R}^{|\mathcal{S}|\times|\mathcal{S}||\mathcal{A}|}$ has rows orthogonal to $\gamma PV^\pi + \mathbf{r}$.*

**Lemma 2.** *Let $\Pi_1$ and $\Pi_2$ be the (optimal) policy matrices for task 1 and task 2, respectively, and let $A \in \mathbb{R}^{|\mathcal{S}|\times|\mathcal{S}|}$ satisfy $AV_1 = V_2$. Then there exists $B \in \mathbb{R}^{|\mathcal{S}||\mathcal{A}|\times|\mathcal{S}||\mathcal{A}|}$ such that*

$$A\Pi_1 B = \Pi_2 \tag{2}$$

Given $\Pi_1$, we can thus obtain $\Pi_2$ by solving for appropriate matrices $A$ and $B$. To relate Lemma 2 to our main claim, we adopt the following restrictive assumption.

**Assumption 1.** *$A$ is a permutation matrix (i.e., the two MDPs are isomorphic up to a permutation of the state space).*

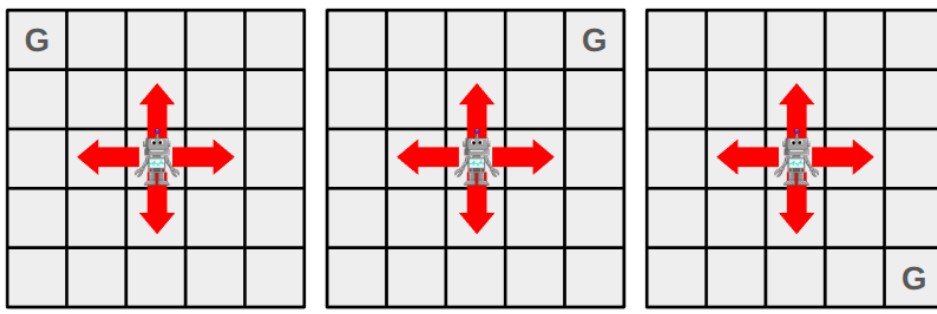

**G**: goal (+ 1 reward)

Figure 2: Toy example demonstrating MDPs that are isomorphic under state permutation.

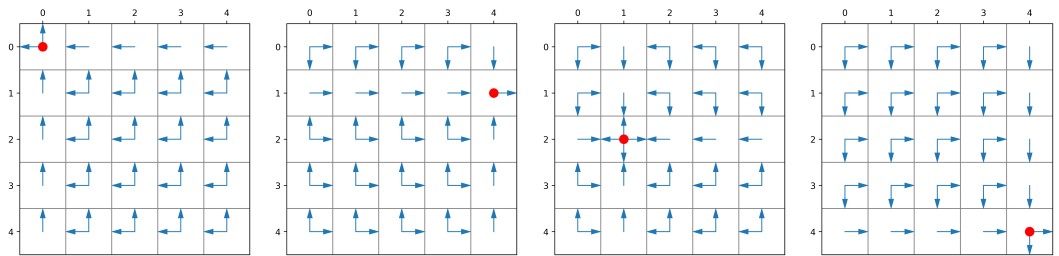

(a) Policy plot extracted from matrix $\Pi_1$

(b) Policy plot extracted from matrix $A_2\Pi_1 B_2$.

(c) Policy plot extracted from matrix $A_3\Pi_1 B_3$

(d) Policy plot extracted from matrix $A_4\Pi_1 B_4$

Figure 3: Policy plots for different goal positions depicted by a red dot. An agent choose one of the four actions(up, down, left, right) and when face the wall, it stay. $A_i$ and $B_i$ represent matrix introduced in equation 16

An illustrative toy example is provided in Figure 2.

**Theorem 1.** *Under Assumption 1, if each state $s$ is represented as a one-hot* row *vector, then the (optimal) policy for task 2 can be written as*

$$\pi_2(\cdot \mid s) = h\Big(\pi_1\big(\cdot \mid g(s)\big)\Big) \tag{3}$$

*where $g : \mathbb{R}^{|\mathcal{S}|} \to \mathbb{R}^{|\mathcal{S}|}$ is linear (e.g., $g(s) = sA$) and $h : \mathbb{R}^{|\mathcal{A}|} \to \mathbb{R}^{|\mathcal{A}|}$ is a (possibly state-dependent) linear map.*

Although such isomorphic MDPs are rare in practice, we later show empirically that our method remains effective even beyond this restrictive case.

We parameterize the policy as a composition of linear and nonlinear maps:

$$\pi = f_{\text{linear}}^{\text{out}} \circ f \circ f_{\text{linear}}^{\text{in}} \tag{4}$$

where $f$ denotes a nonlinear backbone, and $f_{\text{linear}}^{\text{in}}/f_{\text{linear}}^{\text{out}}$ are input/output linear transforms. Because linear maps compose associatively, adding linear maps before and after $\pi$ is equivalent to adding linear maps immediately before and after the backbone. We henceforth *absorb* $f_{\text{linear}}^{\text{out}}$ into $h$ and $f_{\text{linear}}^{\text{in}}$ into $g$, and write

$$h \circ \pi \circ g = \big(h \circ f_{\text{linear}}^{\text{out}}\big) \circ f \circ \big(f_{\text{linear}}^{\text{in}} \circ g\big) =: h \circ f \circ g \tag{5}$$

### 4.2 Analysis of Meta-Training Task Coverage and Adaptation Error

We extend our analysis to evaluate adaptation performance on OOD tasks. Let $\Gamma_{\text{train}}$ denote the set of linear transforms (pre-backbone maps) associated with the meta-training tasks. We write

$$\bar{\mathcal{G}} := \{\, g(s) : \, g \in \Gamma_{\text{train}}, \, s \in \mathcal{S}_{\text{meta}} \,\}$$

for the set of all transformed states observed during meta-training, where $\mathcal{S}_{\text{meta}}$ is the set of states visited under the meta-training policies $\{\pi_t\}_{t=1}^N$ (e.g., the union of the supports of their state-visitation distributions). According to Theorem 1, for any policy backbone $f$, learning task-specific linear layers $g_t$ (pre-backbone) and $h_t$ (post-backbone) yields a policy for a new task $t$:

$$\pi_t(\cdot \mid s) \;=\; h_t\big(f\big(g_t(s)\big)\big). \tag{6}$$

However, meta-training yields an approximate backbone $f_{\text{meta}}$ that is fitted only to a set of transformed states $g(s) \in \bar{\mathcal{G}}$ and is, in general, not reliable over the entire feature space $\mathcal{G}$. In particular,

$$f^*\big(g(s)\big) \;\approx\; f_{\text{meta}}\big(g(s)\big) \quad \forall g(s) \in \bar{\mathcal{G}}, \tag{7}$$

where $f^*$ denotes the ideal backbone that is reliable over $\mathcal{G}$. We assume that, as the diversity of training tasks increases, $\bar{\mathcal{G}} \to \mathcal{G}$. For a test task $t$, let $\mathcal{X}_t := \{ g_t(s) : s \in \mathcal{S} \}$ be the set of transformed inputs encountered at adaptation time, and let $\mathcal{X}_t^{\text{OOD}} := \mathcal{X}_t \setminus \bar{\mathcal{G}}$. We define the adaptation error as

$$\sum_{x \in \mathcal{X}_t} \big\| h_t\big(f^*(x)\big) - h_t\big(f_{\text{meta}}(x)\big) \big\|, \qquad x = g_t(s), \tag{8}$$

where $\| \cdot \|$ denotes a vector norm (e.g., the $\ell_2$ norm).

**Theorem 2.** *Suppose both backbone networks $f_{meta}$ and $f^*$ are L-Lipschitz. Then, the adaptation error is bounded as follows:*

$$\sum_{x \in \mathcal{X}_t} \big\| h_t\big(f^*(x)\big) - h_t\big(f_{meta}(x)\big) \big\| \;\leq\; 2L \|h_t\|_{\text{op}} \cdot |\mathcal{X}_t^{\text{OOD}}| \cdot \epsilon_{max} \tag{9}$$

*where*

- *$\|h_t\|_{\text{op}}$ is the operator norm (e.g., spectral norm) of the linear map $h_t$,*

- *$\mathcal{X}_t^{\text{OOD}} = \{ x \in \mathcal{X}_t : x \notin \bar{\mathcal{G}} \}$ is the set of test-time inputs not encountered during meta-training,*

- *$\epsilon_{max} = \max\limits_{x \in \mathcal{X}_t^{\text{OOD}}} \min\limits_{z \in \bar{\mathcal{G}}} \|x - z\|$ denotes the maximum distance from an OOD input to the meta-training support.*

This bound implies that the adaptation error is controlled by both the number of OOD inputs and their distance from the meta-training support, scaled by the Lipschitz constant of the backbone and the operator norm of the head. As the diversity of meta-training tasks increases, the coverage of $\bar{\mathcal{G}}$ grows, thus reducing both $|\mathcal{X}_t^{\text{OOD}}|$ and the typical values of $\epsilon_{\text{max}}$, which in turn tightens the adaptation error bound for novel tasks.

**Remark.** *Interestingly, we observe that a randomly initialized backbone often induces* near-optimal *policies on meaningful tasks via suitable linear pre-/post-mappings. A plausible explanation is that a randomly initialized policy may already be (near-)optimal for certain degenerate behavioral tasks; consequently, by Theorem 1, such random initializations can be treated as* backbone priors *for other tasks. Detailed results are demonstrated in Section A.5.*

## 5 EXPERIMENTS

### 5.1 EXPERIMENTAL SETUP

We conduct experiments in the MuJoCo control suite (Todorov et al., 2012) with specific tasks such as reaching goals, or achieving target velocities or directions, which are widely used meta-RL benchmark tasks adopted in previous studies (Finn et al., 2017; Zintgraf et al., 2019a; Raghu et al., 2019; Song et al., 2019; Rakelly et al., 2019; Zintgraf et al., 2019b; Beukman et al., 2023; Xu et al., 2022; Wang et al., 2024). While our primary focus is task variation induced by reward functions, we also evaluate the algorithm under variations in transition dynamics. In contrast to most prior approaches that assess adaptation primarily on in-distribution tasks, our method is designed to enhance adaptability to OOD tasks, thereby broadening the practical applicability of meta-learning. We mainly adopt the code used in Rakelly et al. (2019) with some modifications to implement OOD test protocols. Detailed task specifications are provided in Section A.2.

---

**Algorithm 1** Adaptive Policy Backbone (APB)

▷ **Meta-training**
Sample $n$ meta-training tasks $\{\mathcal{T}_1, \ldots, \mathcal{T}_n\} \sim p(\mathcal{T})$
Initialize policy: shared backbone parameters $\psi$, head parameters $\{\rho_i^h\}_{i=1}^n$, tail parameters $\{\rho_i^t\}_{i=1}^n$
Initialize Q-functions $\{\omega_i\}_{i=1}^n$
**while** not converged **do**
   **for** $i = 1$ to $n$ **do**
      Roll out episode in $\mathcal{T}_i$, store transitions in $\mathcal{B}_i$
      Compute critic loss $\mathcal{L}_i^{\text{critic}}$ and actor loss $\mathcal{L}_i^{\text{actor}}$
      Update critic: $\omega_i \leftarrow \omega_i - \eta \nabla_{\omega_i} \mathcal{L}_i^{\text{critic}}$
      Update actor head/tail: $(\rho_i^h, \rho_i^t) \leftarrow (\rho_i^h, \rho_i^t) - \eta \nabla_{(\rho_i^h, \rho_i^t)} \mathcal{L}_i^{\text{actor}}$
   **end for**
   Update backbone: $\psi \leftarrow \psi - \eta \nabla_\psi \left( \frac{1}{n} \sum_{i=1}^n \mathcal{L}_i^{\text{actor}} \right)$
**end while**

▷ **Meta-testing**
Choose OOD task $\mathcal{T}_o \not\sim p(\mathcal{T})$
Initialize backbone $\psi$ with pretrained parameters from meta-training (frozen)
Initialize head $\rho^h$, tail $\rho^t$, Q-function $\omega$
**while** not converged **do**
   **for** $i = 1$ to $M$ **do**
      Roll out episode with $\{\rho^h, \psi, \rho^t\}$ in $\mathcal{T}_o$, store transitions in $\mathcal{B}$
   **end for**
   Compute critic loss $\mathcal{L}^{\text{critic}}$ and actor loss $\mathcal{L}^{\text{actor}}$
   Update critic: $\omega \leftarrow \omega - \eta \nabla_\omega \mathcal{L}^{\text{critic}}$
   Update actor head/tail: $(\rho^h, \rho^t) \leftarrow (\rho^h, \rho^t) - \eta \nabla_{(\rho^h, \rho^t)} \mathcal{L}^{\text{actor}}$
**end while**

---

We meta-train a shared backbone across meta-training tasks until convergence and, during OOD adaptation, freeze it while adapting task-specific linear layers, using a standard TD3 policy module (Fujimoto et al., 2018). We provide a pseudocode for APB in Algorithm 1, in which $M$ and $\eta$ denote the number of trajectories to obtain samples and the learning rate. All hyperparameters used in the experiments are listed in Section A.3.

For the experiments comparing with meta-RL baselines, we evaluate our method against two well-established paradigms: (i) parameterized policy-gradient (PPG) methods that adapt via gradient-based fine-tuning of the parameters (MAML, ANIL, CAVIA; (Finn et al., 2017; Raghu et al., 2019; Zintgraf et al., 2019a)) and (ii) black-box methods that adapt by *online inference*—they *continually update* a task-latent from the incoming context while keeping network weights fixed at test time (PEARL, VariBAD; (Rakelly et al., 2019; Zintgraf et al., 2019b)). We also include the recent transformer-based Meta-DT (Wang et al., 2024).

The experiments are designed to answer the following questions.

- Does APB improve sample efficiency over a standard RL algorithm?
- Does APB achieve superior adaptation performance on OOD tasks compared with existing meta-RL baselines?
- Does meta-trained policy backbone generalize to OOD tasks?

### 5.2 EXPERIMENTAL RESULTS

**Improvement of sample efficiency over a standard RL algorithm.** One particular property of the meta-RL algorithm is to learn new tasks better than standard RL algorithms (Beck et al., 2023).

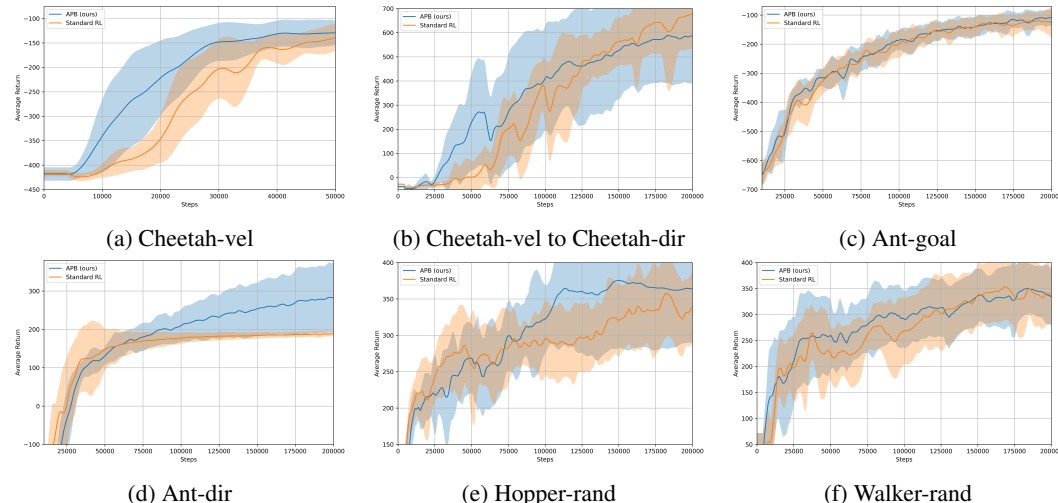

(a) Cheetah-vel     (b) Cheetah-vel to Cheetah-dir     (c) Ant-goal

(d) Ant-dir     (e) Hopper-rand     (f) Walker-rand

Figure 4: Experimental result comparing APB and the standard RL algorithm on the out-of-distribution tasks. Each curve represents the average return over 10 random seeds, with the shaded area indicating one standard deviation from the mean.

For this purpose, meta-RL learns priors from many tasks and leverages them to promote the learning process. In our experiment, we compare APB against a strong standard RL baseline TD3 under the same network capacity for fair comparison. Across most tasks, the proposed method achieves (marginally) better performance than a standard RL algorithm, exhibiting faster convergence and/or higher asymptotic average return for the same number of interactions. The results are demonstrated in Figure 4 and detailed implementations are presented in Section A.6.

**Superior adaptation performance compared with well-established Meta-RL methods.** Despite fine-tuning parameters during adaptation, MAML and CAVIA show a negligible increase in average return. ANIL also fine-tunes policy parameters but freezes the backbone during adaptation, similarly to our proposed method. Black-box algorithms such as PEARL, VariBAD, and Meta-DT fail to adapt under reward variation, whereas they show only slight yet consistent gains under transition variation. We hypothesize that this gap arises because transition variation induces only a mild shift in the observation distribution (task semantics and rewards remain largely unchanged), whereas reward variation changes the objective and drives policies into different regions of the state space, yielding OOD observations that the context encoder was not trained to encode. Across all tasks, our method shows consistent adaptation performance. The results are depicted in Figure 5.

**Behavior cloning.** To assess whether a pretrained policy structure genuinely generalizes to OOD tasks, we perform behavior cloning (BC) (Pomerleau, 1988) on (near-)expert demonstrations collected from OOD tasks. Prior work reports that imitation using expert data often fails to generalize behavior because expert demonstrations cover only a narrow portion of the state–action space (Ross & Bagnell, 2010); accordingly, we deliberately use demonstrations with limited coverage. We train two policies via supervised learning on the same demonstrations: (i) a model that freezes the meta-trained backbone and updates only the task-specific linear layers, and (ii) a randomly initialized model with all parameters unfrozen and updated during training. At evaluation, we roll out each policy on episodes whose horizon exceeds that of the demonstrations ($H_{\text{eval}} > H_{\text{demo}}$), forcing extrapolation beyond the support of the demonstrations and thereby stressing OOD generalization. As shown in Figure 6, the pretrained backbone improves generalization over the baseline on most tasks. Details of the experimental setup are provided in Section A.4.

## 6 DISCUSSION

In this paper, we propose APB, a meta-transfer RL method for OOD task adaptation. Our main claims are twofold: (i) updating only linear layers placed before and after a shared backbone—while

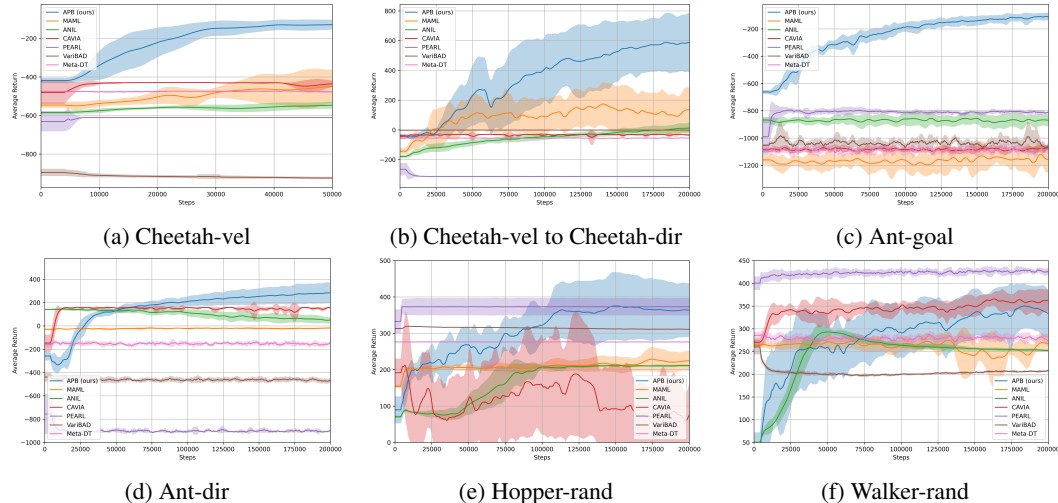

Figure 5: Experimental result comparing APB and the existing meta-RL baselines on the out-of-distribution tasks. Each curve represents the average return over 10 random seeds, with the shaded area indicating one standard deviation from the mean.

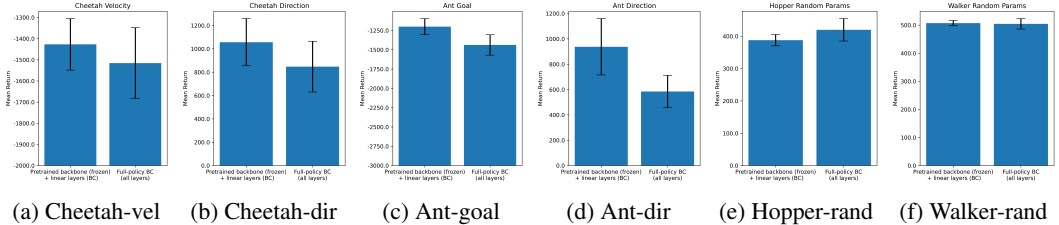

Figure 6: Experimental results on BC. Mean episodic return with 95% confidence intervals across 10 random seeds.

keeping the backbone fixed—suffices to adapt to new tasks, and (ii) APB is capable of generalizing behavior on OOD tasks. We support these claims with a simple theoretical analysis and empirical results on widely used meta-RL benchmarks.

**Limitations.** APB exhibits potential but also has several limitations. First, this study is limited to state-based observations; extending APB to pixel-based observations (e.g., images) remains important future work. Second, although APB requires fewer trainable parameters than standard RL, it does not yield a significant improvement in sample efficiency; further investigation into adaptation/optimization schemes to accelerate learning is a worthwhile direction. Third, the current policy parameterization (fixed backbone with linear pre-/post-layers) may be limited in expressiveness as the number and diversity of tasks grow, implying a capacity–coverage trade-off. We expect that more expressive architectures such as diffusion models (Ho et al., 2020) or transformers (Vaswani et al., 2017) could address these limitations in future work.

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

# A APPENDIX

## A.1 PROOF

### A.1.1 PROOF OF LEMMA 1

**Lemma 1.** *If $V^\pi = (I_{|\mathcal{S}|} - \gamma P^\pi)^{-1} R^\pi$ holds, then using $P^\pi = \Pi P$ and $R^\pi = \Pi \mathbf{r}$ we have $V^\pi = \Pi(\gamma P V^\pi + \mathbf{r})$. Consequently, the policy matrix $\Pi$ can be expressed as*

$$\Pi = \frac{V^\pi (\gamma P V^\pi + \mathbf{r})^\top}{\|\gamma P V^\pi + \mathbf{r}\|^2} + N \quad with \quad N(\gamma P V^\pi + \mathbf{r}) = \mathbf{0}_{|\mathcal{S}|} \tag{10}$$

*where $N \in \mathbb{R}^{|\mathcal{S}| \times |\mathcal{S}||\mathcal{A}|}$ has rows orthogonal to $\gamma P V^\pi + \mathbf{r}$.*

*Proof.* From the definition,

$$V^\pi = (I_{|\mathcal{S}|} - \gamma P^\pi)^{-1} R^\pi = (I_{|\mathcal{S}|} - \gamma \Pi P)^{-1} \Pi \boldsymbol{r} \tag{11}$$

$$\rightarrow (I_{|\mathcal{S}|} - \gamma \Pi P) V^\pi = \Pi \boldsymbol{r} \tag{12}$$

$$V^\pi - \gamma \Pi P V^\pi = \Pi \boldsymbol{r} \tag{13}$$

$$\Pi(\boldsymbol{r} + \gamma P V^\pi) = V^\pi \tag{14}$$

Equation equation 14 represents a linear system, which admits a general solution and a particular solution. Therefore,

$$\Pi = \frac{V^\pi (\gamma P V^\pi + \boldsymbol{r})^\top}{\|\gamma P V^\pi + \boldsymbol{r}\|^2} + N, \quad \text{such that} \quad N(\gamma P V^\pi + \boldsymbol{r}) = \mathbf{0}_{|\mathcal{S}|} \tag{15}$$

$\square$

### A.1.2    PROOF OF LEMMA 2

**Lemma 2.** *Let $\Pi_1$ and $\Pi_2$ be the (optimal) policy matrices for task 1 and task 2, respectively, and let $A \in \mathbb{R}^{|\mathcal{S}| \times |\mathcal{S}|}$ satisfy $AV_1 = V_2$. Then there exists $B \in \mathbb{R}^{|\mathcal{S}||\mathcal{A}| \times |\mathcal{S}||\mathcal{A}|}$ such that*

$$A\Pi_1 B = \Pi_2 \tag{16}$$

*Proof.* We start from the decomposition of $\Pi_1$ and $\Pi_2$:

$$\Pi_1 = \frac{V_1 (\gamma PV_1 + r_1)^\top}{\|\gamma PV_1 + r_1\|^2} + N_1, \tag{17}$$

$$\Pi_2 = \frac{V_2 (\gamma PV_2 + r_2)^\top}{\|\gamma PV_2 + r_2\|^2} + N_2 \tag{18}$$

Choose $A$ so that $AV_1 = V_2$, and define

$$B = \frac{(\gamma PV_1 + r_1)(\gamma PV_2 + r_2)^\top}{\|\gamma PV_2 + r_2\|^2} \tag{19}$$

Then

$$A\Pi_1 B = A\frac{V_1 (\gamma PV_1 + r_1)^\top}{\|\gamma PV_1 + r_1\|^2}B + AN_1 B \tag{20}$$

$$= \frac{V_2 (\gamma PV_1 + r_1)^\top}{\|\gamma PV_1 + r_1\|^2}B + AN_1 B \tag{21}$$

$$= \frac{V_2 (\gamma PV_2 + r_2)^\top}{\|\gamma PV_2 + r_2\|^2} + \underbrace{AN_1 \frac{(\gamma PV_1 + r_1)(\gamma PV_2 + r_2)^\top}{\|\gamma PV_2 + r_2\|^2}}_{\text{rows orthogonal to } (\gamma PV_2 + r_2)} \tag{22}$$

For the residual term, right-multiplying by $(\gamma PV_2 + r_2)$ yields

$$AN_1 \frac{(\gamma PV_1 + r_1)(\gamma PV_2 + r_2)^\top}{\|\gamma PV_2 + r_2\|^2}(\gamma PV_2 + r_2) = AN_1(\gamma PV_1 + r_1) \tag{23}$$

$$= A\mathbf{0}_{|\mathcal{S}|} = \mathbf{0}_{|\mathcal{S}|}, \tag{24}$$

so its rows are orthogonal to $(\gamma PV_2 + r_2)$. Hence, setting

$$N_2 := AN_1 \frac{(\gamma PV_1 + r_1)(\gamma PV_2 + r_2)^\top}{\|\gamma PV_2 + r_2\|^2} \tag{25}$$

gives

$$A\Pi_1 B = \frac{V_2 (\gamma PV_2 + r_2)^\top}{\|\gamma PV_2 + r_2\|^2} + N_2 = \Pi_2. \tag{26}$$

□

### A.1.3    PROOF OF THEOREM 1

**Theorem 1.** *Under Assumption 1, if each state $s$ is represented as a one-hot* row *vector, then the (optimal) policy for task 2 can be written as*

$$\pi_2(\cdot \mid s) = h\Big(\pi_1\big(\cdot \mid g(s)\big)\Big) \tag{27}$$

*where $g : \mathbb{R}^{|\mathcal{S}|} \to \mathbb{R}^{|\mathcal{S}|}$ is linear (e.g., $g(s) = sA$) and $h : \mathbb{R}^{|\mathcal{A}|} \to \mathbb{R}^{|\mathcal{A}|}$ is a (possibly state-dependent) linear map.*

*Proof.* Let $\tilde{\pi}(s)$ be the extended form of policy $\tilde{\pi}(s) : \mathbb{R}^{1 \times |\mathcal{S}|} \to \mathbb{R}^{1 \times |\mathcal{S}||\mathcal{A}|}$, where only the entries corresponding to state $s$ are nonzero, representing the action probabilities. All other entries are zero. For example, if $|\mathcal{S}| = 2, |\mathcal{A}| = 2$, and the input state is $s_1$, then the policy network outputs

$$\tilde{\pi}(s_1) = [\pi(a_1|s_1), \pi(a_2|s_1), 0, 0]$$

and for $s_2$, the output is

$$\tilde{\pi}(s_2) = [0, 0, \pi(a_1|s_2), \pi(a_2|s_2)]$$

Let $EXT(\cdot)$ be the policy extraction operator such that $EXT(\tilde{\pi}(s)) = \pi(\cdot|s)$. Since the state is represented as a one-hot vector, we have:

$$s\Pi = \tilde{\pi}(s) \tag{28}$$

Because $\Pi_2 = A\Pi_1 B$, we have

$$\pi_2(\cdot|s_i) = EXT(\tilde{\pi}_2(s_i)) \tag{29}$$
$$= EXT(s_i\Pi_2) \tag{30}$$
$$= EXT(s_i A\Pi_1 B) \tag{31}$$

Write $s_i A = s_k$ (i.e., $A$ maps the row corresponding to $s_i$ to that of $s_k$). Then

$$\pi_2(\cdot|s_i) = EXT(s_k\Pi_1 B) \tag{32}$$
$$= EXT(\tilde{\pi}_1(s_k)B) \tag{33}$$

Since $\tilde{\pi}_1(s_k)$ has support only on the $k$-th state block, $\tilde{\pi}_1(s_k)B$ depends solely on the $k$-th row block of $B$. Hence, regardless of the specific extraction mechanism, applying $EXT$ returns a $|\mathcal{A}|$-dimensional *linear* image of $\pi_1(\cdot \mid s_k)$; that is, there exists a matrix $B' \in \mathbb{R}^{|\mathcal{A}| \times |\mathcal{A}|}$ such that

$$EXT(\tilde{\pi}_1(s_k)B) = \pi_1(\cdot \mid s_k)B' \tag{34}$$

We define $g$ by $g(s_i) = s_k$ (equivalently, $s_i A = s_k$), and define $h$ by $h(\pi(\cdot|s)) = \pi(\cdot|s)B'$. With these definitions, we can express $\pi_2$ as the composition $h \circ \pi_1 \circ g$, which completes the argument. $\square$

### A.1.4 Proof of Theorem 2

**Theorem 2.** *Suppose both backbone networks $f_{meta}$ and $f^*$ are L-Lipschitz. Then, the adaptation error is bounded as follows:*

$$\sum_{x \in \mathcal{X}_t} \left\| h_t(f^*(x)) - h_t(f_{meta}(x)) \right\| \leq 2L\|h_t\|_{op} \cdot |\mathcal{X}_t^{OOD}| \cdot \epsilon_{max} \tag{35}$$

*Proof.*

$$\sum_{g_t(s) \in \mathcal{G}} \left\| h_t\Big(f^*\big(g_t(s)\big)\Big) - h_t\Big(f_{\text{meta}}\big(g_t(s)\big)\Big) \right\| \tag{36}$$

$$= \sum_{g_t(s) \in \mathcal{G}} \left\| h_t\Big(f^*\big(g_t(s)\big) - f_{\text{meta}}\big(g_t(s)\big)\Big) \right\|$$

$$\leq \|h_t\|_{\text{op}} \cdot \sum_{g_t(s) \in \mathcal{G}} \left\| f^*\big(g_t(s)\big) - f_{\text{meta}}\big(g_t(s)\big) \right\| \tag{37}$$

$$\leq \|h_t\|_{\text{op}} \cdot \sum_{g_t(s) \in \mathcal{G}} \Bigg( \underbrace{\left\| f^*\big(g_t(s)\big) - f^*\big(\hat{g}_t(s) + \epsilon(s)\big) \right\|}_{\leq L\|\epsilon(s)\|}$$

$$+ \underbrace{\left\| f^*\big(\hat{g}_t(s) + \epsilon(s)\big) - f_{\text{meta}}\big(\hat{g}_t(s) + \epsilon(s)\big) \right\|}_{=0} \tag{38}$$

$$+ \underbrace{\left\| f_{\text{meta}}\big(\hat{g}_t(s) + \epsilon(s)\big) - f_{\text{meta}}\big(g_t(s)\big) \right\|}_{\leq L\|\epsilon(s)\|} \Bigg)$$

$$\text{where } \hat{g}_t(s) = \arg\min_{g'(s) \in \bar{\mathcal{G}}} \big(\|g_t(s) - g'(s)\|\big), \quad \epsilon(s) = g_t(s) - \hat{g}_t(s)$$

$$\leq 2L\|h_t\|_{\text{op}} \cdot \sum_{g_t(s) \in \bar{\mathcal{G}}^{\complement}} \|\epsilon(s)\| \tag{39}$$

$$\leq 2L\|h_t\|_{\text{op}} \cdot |\bar{\mathcal{G}}^{\complement}| \cdot \epsilon_{\max} \tag{40}$$

$$\square$$

## A.2 TASK DESCRIPTION

Table 1: Task description

|  | Task (objective) | Meta train | Meta test |
|---|---|---|---|
| Reward | **Cheetah-vel** (Reaching velocity $(v)$) | $v \sim \mathcal{U}(0.0, 3.0)$ | $v = -2.0$ |
|  | **Cheetah-vel to Cheetah-dir** (Going target direction) | $v \sim \mathcal{U}(0.0, 3.0)$ | backward |
|  | **Ant-goal** (Reaching goal position $(x, y)$) | $\theta \sim \mathcal{U}(0, \pi)$ $x, y = 3\cos\theta, 3\sin\theta$ | $\theta = 1.5\pi$ $x, y = 3\cos\theta, 3\sin\theta$ |
|  | **Ant-dir** (Going target direction $(\theta)$) | $\theta \sim \mathcal{U}(0, \pi)$ | $\theta = 1.5\pi$ |
| Dynamics | **Hopper-rand** (Going forward) **Walker-rand** (Going forward) | $\alpha \sim \mathcal{U}(-3.0, 2.1)$ $1.5^{\alpha} \times$body mass $1.5^{\alpha} \times$body inertia $1.3^{\alpha} \times$damping | $\alpha = 2.4$ $1.5^{\alpha} \times$body mass $1.5^{\alpha} \times$body inertia $1.3^{\alpha} \times$damping |

**Cheetah-vel task.** The agent is required to reach a target velocity $v$ using a cheetah model. During meta-training, target velocities are uniformly sampled from $\mathcal{U}(0.0, 3.0)$, and in meta-test, the agent is evaluated at $v = -2.0$.

**Cheetah-vel to Cheetah-dir task.** This task is designed to evaluate the generalization from the Cheetah-vel to the Cheetah-dir setting. The agent is trained to move in randomly sampled target velocities $v \sim \mathcal{U}(0.0, 3.0)$ during meta-training and is tested in the backward direction during meta-test.

**Ant-goal task.** The ant agent is required to reach a goal position $(x, y)$ located on a semicircle of radius 3. For meta-training, the target angle $\theta$ is sampled uniformly from $[0, \pi]$ and the corresponding goal is set as $x = 3\cos\theta, y = 3\sin\theta$. In meta-test, the agent is evaluated at $\theta = 1.5\pi$, that is, $x = 3\cos\theta, y = 3\sin\theta$.

**Ant-dir task.** In this task, the ant must move in a specified direction $\theta$. During meta-training, the direction is uniformly sampled from $\mathcal{U}(0, \pi)$. In meta-test, the agent is evaluated at a fixed direction $\theta = 1.5\pi$.

**Hopper-rand task.** The agent is required to move forward while adapting to variations in the dynamics. During meta-training, the body mass, inertia, and damping parameters are randomized according to the factor $\alpha \sim \mathcal{U}(-3.0, 2.1)$, with each parameter scaled as $1.5^\alpha \times$body mass, $1.5^\alpha \times$body inertia, and $1.3^\alpha \times$damping. In meta-test, the agent is evaluated with $\alpha = 2.4$.

**Walker-rand task.** Similar to Hopper-rand task, the walker agent must move forward and adapt to dynamic variations. During meta-training, mass, inertia, and damping are scaled with $\alpha \sim \mathcal{U}(-3.0, 2.1)$, and at meta-test, the parameters are set using $\alpha = 2.4$.

A.3 HYPERPARAMETERS

Table 2: Hyperparameters for adaptation

| Hyperparameter | Value(s) |
| --- | --- |
| Number of trajectories | 10 |
| Parameter noise | 0.00008 (Cheetah-dir); 0.005 (Cheetah-vel); 0.006 (Ant-goal); 0.00075 (Ant-dir); 0.007 (Hopper-rand); 0.03 (Walker-rand) |
| Policy initialization coefficient | 0.07 (Ant-dir); 0.1 (Hopper-rand); 0.01 (Walker-rand); 0.001 (Others) |
| Reset episode | 15 (Cheetah-vel); 25 (Ant-goal); 10 (Others) |
| Number of updates | 1000 (Cheetah-dir, Ant-dir); 2000 (Others) |
| Batch size | 512 |
| Buffer size | 200000 |
| Learning rate | 0.0001 (Actor); 0.05 (Critic) |
| Number of meta-train tasks | 30 |

We provide brief explanations of several hyperparameters below:

- **Number of trajectories**: Number of trajectories sampled before running an update block.

- **Parameter noise**: Standard deviation of the parameter-perturbation noise applied during data collection.

- **Policy initialization coefficient**: Scale factor for initializing the task-specific linear layers; smaller coefficient keeps the layers near zero and stabilizes early adaptation.

- **Reset episodes**: To mitigate early overfitting, we reinitialize the task-specific parameters every $n$ episodes (the "reset" technique (Nikishin et al., 2022)).

- **Number of updates**: Number of gradient update steps performed after each data-collection phase (i.e., after sampling $n_{\text{traj}}$ trajectories).

A.4 BEHAVIOR CLONING

**Demonstrations.** We collect (near-)expert demonstrations $\mathcal{D}_t = \{(s, a)\}_{i=1}^N$ for each *OOD* test task by rolling out a (near-)expert policy $\pi_{\exp}$. Each per-task dataset contains $N = 100{,}000$ state–action pairs (transitions) aggregated across multiple episodes. The demonstrations consist of contiguous expert rollouts with episode horizon $H_{\text{demo}}$ and therefore concentrate mass on the expert visitation distribution $d^{\pi_{\exp}}$ (narrow state–action support).

**Policy parameterizations.**  We evaluate two models trained on the same demonstrations.

- **APB-BC (ours).** The meta-trained backbone $f_{\text{meta}}$ is *frozen*. We learn only the task-specific linear layers $g$ (pre-backbone) and $h$ (post-backbone). Both layers are initialized with scale $w$ from Table 2.

- **Full model (baseline).** Identical architecture but *randomly initialized*; all parameters are unfrozen and updated during training.

**Objective.**  Following standard continuous-control BC, we use a deterministic policy and minimize the mean-squared error between actions and expert actions:

$$\mathcal{L}_{\text{BC}}(\theta) \;=\; \frac{1}{|\mathcal{B}|} \sum_{(s,a)\in\mathcal{B}} \big\| \pi_\theta(s) - a \big\|_2^2,$$

where $\mathcal{B}$ is a mini-batch. (For completeness, a stochastic alternative is the negative log-likelihood $\mathcal{L}_{\text{NLL}}(\theta) = -\frac{1}{|\mathcal{B}|} \sum \log \pi_\theta(a \mid s)$, but we do not use it in our experiments.)

**Optimization and preprocessing.**  We optimize with Adam (PyTorch), a mini-batch size of 2048, using *task-dependent* learning rates:

$$\text{LR} = \begin{cases} 5 \times 10^{-4}, & \text{Walker-rand-params} \\ 1 \times 10^{-4}, & \text{Hopper-rand-params} \\ 1 \times 10^{-3}, & \text{All other tasks (Cheetah-dir, Cheetah-vel, Ant-goal, Ant-dir).} \end{cases} \tag{41}$$

**Evaluation.**  After training, we roll out each policy for episodes with horizon $H_{\text{eval}} > H_{\text{demo}}$ to stress extrapolation beyond the demonstration support (OOD generalization). Performance is measured by episodic return. We report the mean across 10 random seeds and 95% confidence intervals.

### A.5  RANDOM INITIALIZATION FOR BACKBONE

**Motivation.**  As suggested in Theorem 1, linear pre-/post-mappings can transport policies across tasks. Empirically, we find that even a *randomly initialized* backbone can, with suitable linear layers, induce *near-optimal* behavior on some meaningful tasks. Intuitively, a random backbone implements a fixed random feature map; for certain "degenerate" behavioral objectives, a linear head on top of such features can already be (near-)optimal. If so, the random backbone can serve as a *backbone prior* for other tasks via task-specific linear layers.

**Protocol.**  We randomly initialize a backbone, freeze all backbone parameters, and train only the task-specific linear layers $g$ (pre-backbone) and $h$ (post-backbone). We refer to this variant as **APB (random init)**. Unless noted otherwise, we use the same adaptation budgets and per-task hyperparameters as in the main APB experiments (Table 2); layer scales are initialized with the policy initialization coefficient from the table.

**Results.**  Across all evaluated environments, as shown in Figure 7, *APB (random init)* yields competent performance, although it does not attain the optimal policy. Notably, on ANT-GOAL the method approaches near-optimal performance with only a small gap. These findings support our remark that a randomly initialized backbone can function as a useful *backbone prior* when paired with task-specific linear layers.

**Practical takeaway.**  If meta-training is infeasible, *APB (random init)* is a surprisingly strong and cheap baseline: use a sufficiently wide backbone, freeze it, and adapt only linear layers.

**Reproducibility details.**  We use identical optimizers and update schedules as in the APB adaptation runs (per-task learning rates and update counts in Table 2); only the backbone source differs (random vs. meta-trained).

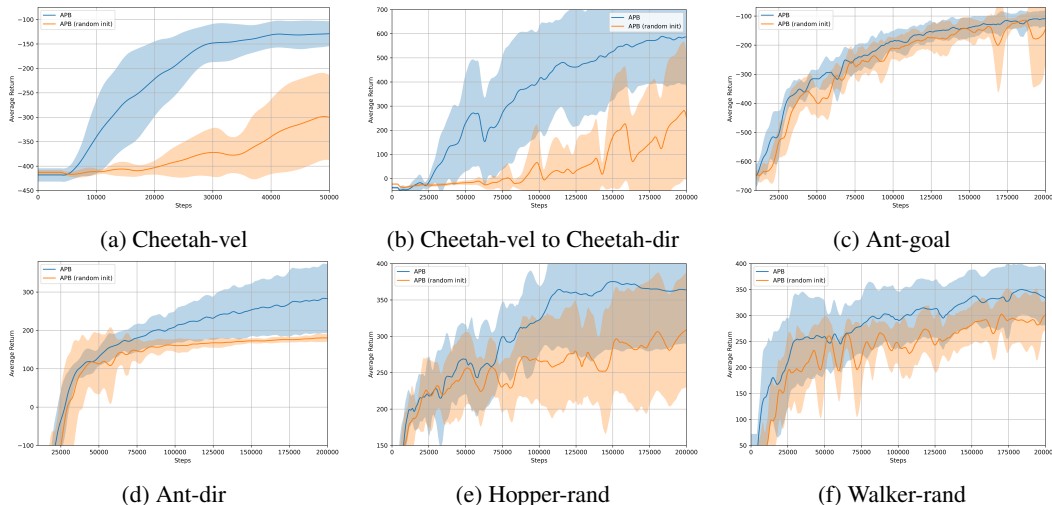

Figure 7: Experimental result comparing APB with pretrained backbone and APB with random initialization on the out-of-distribution tasks. Each curve represents the average return over 10 random seeds, with the shaded area indicating one standard deviation from the mean.

## A.6 PRACTICAL IMPLEMENTATION OF APB

During meta-training, a single policy backbone is shared across all tasks, with task-specific *linear layers* placed before and after the backbone; each task also has its own Q-functions (critics). After meta-training, we freeze the backbone and train new task-specific linear layers on the meta-test task using the same optimization pipeline; unlike meta-training, only this small subset of policy parameters is updated at adaptation time. To encourage sufficiently rich exploration with few trainable parameters, we consider two exploration protocols *separately*: (i) evolutionary-style *parameter-space perturbation* (Plappert et al., 2017), and (ii) *action-space* Gaussian noise in the style of TD3 (noise added to the mean action) (Fujimoto et al., 2018). For a fair comparison with standard RL, both APB and the standard RL baseline are evaluated under *both* exploration variants using identical budgets; unless otherwise noted, Figure 4 reports, for each method, the better-performing variant on each task. To mitigate early overfitting of the task-specific adapters, we also adopt a *reset* strategy that periodically reinitializes these parameters every $n$ episodes (Nikishin et al., 2022); the same schedule is used for APB and the standard RL baseline. Unless otherwise stated, APB and the standard RL baseline share identical training budgets (environment steps, number of updates, batch size, random seeds) and the same evaluation protocol.

## A.7 IMPLEMENTATION OF META-RL BASELINES

We use the authors' public implementations *and* their default hyperparameters for all baselines: MAML (Finn et al., 2017), CAVIA (Zintgraf et al., 2019a), PEARL (Rakelly et al., 2019), VariBAD (Zintgraf et al., 2019b), and Meta-DT (Wang et al., 2024). ANIL (Raghu et al., 2019) is implemented by modifying our MAML code to freeze the feature backbone during adaptation (inner loop), while keeping the head trainable; we otherwise reuse the same meta-training and adaptation hyperparameters as in MAML. For a fair comparison, we keep the policy network size (hidden depth and units per layer) identical across all methods.

