# OpenReview forum: "Adaptive Policy Backbone via Shared Network"
_ICLR.cc/2026/Conference — ICLR 2026 Conference Withdrawn Submission_

### Official Review · Reviewer_QFW1 · 2025-10-27

**Soundness:** 2
**Presentation:** 2
**Contribution:** 2
**Rating:** 4
**Confidence:** 2

**Summary:**

To solve the out-of-distribution adaptation problem, the authors have proposed a new meta learning approach. In the proposed approach, a novel policy structure is designed called adaptive policy backbone (APB). With this policy structure, a parameter-efficient finetuning can be achieved. The proposed approach is evaluated in the mujoco domain.

**Strengths:**

1.	The proposed approach is not only empirically evaluated, but also well theoretically grounded.

2.	The paper is well written, so the importance of OOD adaptation is adequately conveyed.

**Weaknesses:**

1.	It is unclear how different the structure in Figure 1 is from the common multi-head policy structure, which is widely employed in the multi-task learning setting. This is related to the novelty of the proposed approach.

2.	As APB is nearly the same as the multi-head policy network, how can it achieve OOD adaptation? Is it related to the freezing learning scheme in the meta-testing phase?

3.	The experiments have shown that the proposed approach can work in certain OOD settings. However, these settings have not demonstrated significant difference between meta-training and meta-testing. Can the proposed approach adapt to more difference between meta-training and meta-testing? For example, the simulated robots have different morphologies.

**Questions:**

Please see the weaknesses part.

---

### Official Review · Reviewer_LC2o · 2025-10-31

**Soundness:** 3
**Presentation:** 3
**Contribution:** 2
**Rating:** 4
**Confidence:** 4

**Summary:**

The paper proposes Adaptive Policy Backbone (APB), a meta-transfer RL method designed to improve sample efficiency and adaptation to out-of-distribution (OOD) tasks. APB introduces a shared policy backbone with lightweight linear pre- and post-layers, enabling parameter-efficient fine-tuning while preserving prior knowledge. Theoretical analyses support the claim that updating only these linear layers can adapt to new tasks, and empirical results on MuJoCo benchmarks demonstrate improved OOD adaptation in terms of mean return compared with existing meta-RL baselines.

However, the theoretical analysis is overly restrictive, and the empirical performance gains are not particularly strong.

**Strengths:**

The insertion of lightweight linear layers before and after a shared backbone offers a clean and parameter-efficient approach to meta-transfer RL, bridging ideas from PEFT and meta-learning. The algorithm design is intuitive and easy to implement.

The authors did comprehensive experiments on six MuJoCo environments cover both reward and dynamics shifts.

**Weaknesses:**

1. The core theorem relies on isomorphic MDPs (state permutation assumption), which rarely hold in practice. The theoretical link between this case and realistic OOD adaptation remains somewhat heuristic.
2. Although APB reduces the number of trainable parameters, the reported gains in sample efficiency over standard RL are modest, suggesting limited practical advantage. Moreover, APB seems to induce higher variance than standard RL and baseline meta RL algorithms.

**Questions:**

1. Meta-RL typically assumes differences in transition dynamics (and possibly rewards) across environments. This work focuses only on reward shifts in both the problem formulation and theoretical analysis. Could the authors justify this simplification or discuss its implications for broader generalization?
2. The experimental improvements are relatively weak. The gains over standard RL are modest, and APB is not consistently better than other meta-RL baselines. Since PEFT methods primarily aim for computational efficiency, it would strengthen the paper if the authors could provide quantitative evidence of reduced computational cost or training time compared to existing baselines.s is overly restrictive, and the empirical performance gains are not particularly strong.

---

### Official Review · Reviewer_Z8P4 · 2025-11-01

**Soundness:** 1
**Presentation:** 4
**Contribution:** 1
**Rating:** 2
**Confidence:** 4

**Summary:**

Authors propose Adaptive Policy Backbone, a method that does Meta RL adaptation by only tuning two task-specific linear layers (on the inputs, and on the outputs).

**Strengths:**

The proposed method is very straightforward, and the paper is written extremely clearly.

**Weaknesses:**

- “Across most tasks, the proposed method achieves (marginally) better performance than a standard RL algorithm, exhibiting faster convergence and/or higher asymptotic average return for the same number of interactions. The results are demonstrated in Figure 4” Figure 4 does not show better performance for APB than for the standard RL algorithm, neither with faster convergence nor asymptotic average return, across most tasks. Asymptotically, only (d) shows statistically significant differences between the methods. With respect to faster convergence, only (a). Please correct me if I am missing something, but this does not seem like an honest reading of the numbers being reported.
- The algorithm formulation is incomplete. Critic and actor losses are undefined.
- The core idea of the project is extremely limited. For some given pre-trained policy, there is only a limited set of new tasks for which two additional linear layers would suffice for adaptation. The toy example for theoretical study is just not useful.
- From the abstract: “Our results show that APB improves sample efficiency over standard RL”. From the limitations statement: “it does not yield a significant improvement in sample efficiency;”.

**Questions:**

“Furthermore, it is well established that fine-tuning only a subset of parameters can significantly improve sample efficiency and reduce training costs” Can you please add citations for this?

---

### Note · Authors · 2025-11-17

I have read and agree with the venue's withdrawal policy on behalf of myself and my co-authors.